# Methods of Definitive Correction of Mandibular Deformity in Hemimandibular Hyperplasia Based on Radiological, Anatomical, and Topographical Measurements—Proposition of Author’s Own Protocol

**DOI:** 10.3390/ijerph191610005

**Published:** 2022-08-13

**Authors:** Kamil Nelke, Klaudiusz Łuczak, Maciej Janeczek, Edyta Pasicka, Monika Morawska-Kochman, Maciej Guziński, Maciej Dobrzyński

**Affiliations:** 1Practice of Maxillo-Facial Surgery and Maxillo-Facial Surgery Ward, EMC Hospital, Pilczycka 144, 54-144 Wrocław, Poland; 2Department of Biostructure and Animal Physiology, Wrocław University of Environmental and Life Sciences, Kożuchowska 1, 51-631 Wrocław, Poland; 3Department of Head and Neck Surgery, Otolaryngology, Wrocław Medical University, Borowska 213, 50-556 Wrocław, Poland; 4Department of Radiology, Wrocław Medical University, Borowska 213, 50-556 Wrocław, Poland; 5Department of Pediatric Dentistry and Preclinical Dentistry, Wrocław Medical University, Krakowska 26, 50-425 Wrocław, Poland

**Keywords:** condylar hyperplasia, hemimandibular hyperplasia, overgrowth correction, asymmetry, mandible

## Abstract

In order to fully evaluate and establish the degree of bone overgrowth, various radiological studies are essential in the careful planning of the amount of surgical excision. In the presented paper, the authors use self-designed anatomo-topographical reference points for planning the surgeries. Routine panoramic radiographs and low-dose computed tomography based on anatomical landmarks help in measuring the proportions of mandibular bone overgrowth with the following preoperative anatomical landmarks: (Go-Go), (Go(Right)-Gn), (Go(Left)-Gn), and (Me–Gn). Measurements taken at selected points and landmarks (gonion-gnathion/gnathion-menton) are easy to conduct. In the authors’ proposal, the main key factor is total chin correction, which is necessary in cases of severe overgrowth; when F0 > C and Go-Gn>, there is >7 mm of vertical bone overgrowth, and the mandibular canal is positioned <5 mm from the inferior mandibular border—MIB. Larger overgrowths (>7 mm) have a greater outcome on the final symmetry than smaller overgrowths. As no guidelines are known, the authors present their own proposal.

## 1. Introduction

Surgery is required for mandibular abnormalities, which result in facial asymmetry, a lack of balance, inappropriate mandibular proportions, and/or decreased jaw, speech, or masticatory functions. Most cases are treated with various orthognathic procedures. Presented herein, condylar hyperplasia (HH/CH2)-related cases at first require the removal of part of the affected condyle, causing an abnormal one-sided mandibular overgrowth. Secondly, the degree of undesirable mandibular overgrowth can be treated in one of various ways and modifications [1,2,3]. Achieving proper symmetry, or at least near-normal symmetrical dimensions of the inferior mandibular border (MIB), is considered to be an excellent surgical outcome. In the presented paper, the authors’ experience and proposal on a definitive mandibular symmetry correction in mandibular hyperplasia are presented and discussed.

Condylar hyperplasia is related to one-sided abnormal pathological overgrowth of the mandible, related to a condyle head increase and prolonged growth in time [1,2,3,4,5,6,7,8]. Its etiology remains not fully known, while occurrence time, gender-related factors, growth components, and severity have been widely discussed in the known literature [1,7,9,10,11,12,13,14,15,16,17,18]. Olate et al.’s and Nolte et al.’s papers suggest more occurrence in the left condyle, with slight female predominance, while growth activity in time influences a later degree of mandibular asymmetry and also even secondary maxillary bite plane deviation [7,9,15,16]. The presented asymmetry was first described by Obwegeser and Makek, and later updated by Wolford et al.; however, few types and forms of the described pathology causing visible mandibular asymmetry are also known [2,3,17,18,19,20].

Many authors have described various techniques and strategies for the treatment of any case of condylar head hyperplasia that results in mandibular asymmetry [4,5,6,7,21]. A condylectomy, in one of its various forms and modifications, is the gold standard for treating the active growth type of asymmetry [8,9,10,15,19,20,21,22,23,24]. The main aspect of a condylectomy focuses on the removal of a part of the mandibular condyle head responsible for mandibular overgrowth and pathological growth present after maturation or severe cases of overgrowth in children. In most cases, an orthognathic procedure (consisting of LeFort I and BSSO—bilateral sagittal split osteotomy) with or without corrective genioplasty procedures is enough to restore a balanced and symmetrical profile [11,14,19,20,21,22,23,24,25]. Some extensive overgrowths require some additional surgical steps. Such cases have some characteristic features such as overgrowth of the mandible on the affected side, a descending and low-positioned mandibular canal with inferior alveolar nerve, a visible lack of bone symmetry, increased vertical ramus height, and enlargement of an entire one-sided mandibular body [26,27,28,29]. It is quite often the case that, in order to achieve a balanced profile, surgery involving inferior alveolar nerve (IAN) repositioning might be troublesome from a surgical point of view, and/or with patients who are unwilling to undergo any additional surgeries in close proximity to the neurovascular system because of the potential for injury of IAN [12,13,14,29,30,31,32]. In most cases, it is challenging to achieve excellent symmetry with a balanced profile or a level of symmetry that is close to the individual’s value.

At first, if any abnormal pathological growth is present in the mandibular condyle, detailed diagnostics consisting of RTG (panx-panoramic radiograph and ceph-lateral cephalogram), CT/LDCT (computed tomography/low-dose computed tomography), and SPECT (single-photon-emission computed tomography) evaluation are necessary [24,28,30]. In most cases, a prolonged growth and difference in bone SPECT uptake ranging from 10 to 15% are present in at least two measurements and are sufficient to qualify any patient for a condylectomy procedure [1,7,8,30,31,32]. SPECT is truly sensitive but not specific for condylar hyperplasia, because there are several types of diseases connected with increased radio uptake, e.g., inflammations, tumors, and joint overload [5,6,7,8,24,30]. The mentioned condylectomy procedure is the first and most important aspect of the treatment of any case of condylar hyperplasia [1,2,3,4,5,29,32].

Condylectomy as the sole procedure can change the position of the mandible, as the mandible auto-rotates after part of the affected condyle is resected [1,2,3,4,5,6,7,8,9,10]. A slight mandibular rotation toward the excised side minimizes the excess growth; however, this movement strongly coincides with the occurrence of a contra-lateral open bite, which enables it by reducing the gap between the glenoid fossa and the neo-head from the bony stump [5,6,7,8]. The following open bite and the degree of malocclusion influence the type of surgical approach, including the decision to proceed in stages (step-by-step procedure) [3,4,5,6,7,8,9,10,11,12,13,14,15] or with an all-in-one approach (one surgical approach) and rarely a surgery-first approach [7,8,9,10,22,23,24]. If the degree of an open bite is limited, and the condylar head is significantly enlarged in three dimensions, in some cases, a mandibular osteotomy should be combined with a condylectomy to improve the future neo-condyle position in the fossa and to decrease its instability and the probability of luxation [3,4,5,6,7,10,11,12,13].

Secondly, each mandibular bone overgrowth has its own individual volume but similar clinical and radiological features that are difficult to mislead from any other mandibular asymmetries [1,2,12,13,14,15,16,24]. In cases of severe overgrowths and troublesome asymmetries, the standard surgical protocol includes one of the used orthognathic surgery procedures [18,19,20,21,22,23]. Despite good correction of jaw proportions and occlusion, various degrees of mandibular overgrowth might still be a problem and those require some special attention from clinicians.

Nevertheless, the scope of surgical intervention for total mandibular asymmetry correction remains greatly individual mostly because of the degree of overgrowth, volume of enlarged bone, patients’ requirements, used approaches/surgical steps, and their timing. A lot of clinical modalities and possible surgical steps are known: (1) condylectomy alone/proportional condylectomy and then orthognathic surgery after orthodontic preparation; (2) condylectomy with surgery-first approach and later orthodontics; (3) condylectomy with/without genioplasty/wing genioplasty procedure with orthognathic surgery (e.g., Ferguson-technique-BSSO-modification) all in one or (4) performed later after 6–12 months; (5) condylectomy alone as sole procedure (in low-progressed bone asymmetries and overgrowths); (6) condylectomy and early/later corrective surgery (bone remodeling and genioplasty); (7) late surgery only for MIB correction such as genioplasty, anguloplasty, and overgrowth bone reshaping (drilling, chiseling, and marginectomy); (8) secondary surgery for definitive correction of mandibular asymmetry after previous condylectomy or when pathological growth is absent/ceased in time. All of the mentioned techniques, their timing, and used approaches are greatly individually established based on surgeons’ preferences and patients’ needs (esthetic, functional, symmetry, and others), and so far, no adequate standardization of treatment is known [3,4,5,6,7,8,22,23,24,25,26,27,28,29,30,31,32,33].

The authors believe that as no adequate guidelines and indications for total mandibular symmetry correction have been established, it is necessary to create a protocol to increase the surgical point of knowledge on the following topic. The presented and used anatomical landmarks in the following prospective study will perhaps improve surgeons’ planning and overall concept. This study aims to present a radiological protocol that defines a set of approaches for mandibular asymmetry correction in a prospective study.

## 2. Materials and Methods

After careful data selection from 2012 to 2022 of all patients from the author’s data-set treated because of various forms of skeletal asymmetry, just a small group of them required some advanced surgical interventions for a total mandibular body correction. In detailed preclinical planning based on LDCT or panoramic radiographs, a total of 12 people (10 females and 2 males; *p* > 0.05; min = 19; max = 33; *p* > 0.05) had skeletal asymmetry and unilateral mandibular bone overgrowth of active or inactive hemimandibular hyperplasia and/or condylar hyperplasia type 2 etiology (HH/CH2), which required additional surgical steps. There was a notable 5:1 ratio of females to males, but no significant statistical correlations between gender, age, or the type of additional surgery were found. Likewise, there were no correlations regarding etiological factors causing the HH/CH2 other than those possibly known [1,2,15,16]. A proposal of a surgical protocol and its usage in various approaches, along with an attempt to describe the usage of the given methods, is presented and discussed based on the preclinical radiological data collected (Table 1; Figure 1). Perhaps the presented prospective study based on anatomic reference points will highlight a new surgical proposal.

This study followed the Declaration of Helsinki on medical protocol and ethics and the Regional Ethical Review Board of the local Medical University approved the study.

According to the available knowledge [1,2,3,4,5,6,7,8,9,10] and the authors’ own proposal, the following points were adopted as the criteria for qualifying the patient for the mandibular asymmetry correction surgery procedure:(1)Patients with active growth or growth cessation in the pathological (hyperplastic) condyle, confirmed by at least two scintigraphic measurements taken at least six months apart;(2)Patients who required condylectomy as the sole procedure and/or performed in various modalities and/or just observation without joint surgery;(3)Multiple degrees (low, mild, or severe) of mandibular asymmetry and overgrowths caused by the hyperplastic condyle (active or previous growth);(4)Patients treated, diagnosed, and/or operated on only by the authors;(5)Full clinical and radiological data standardized and performed with the same technique, the same person, and following the authors’ protocol;(6)Patient cases with mandibular asymmetrical overgrowth caused by present or past growth in the affected condyle;(7)Other mandible asymmetries caused by HH/CH2, or other factors related to recent growth.

It was assumed that the contraindications based on the gathered data and the authors’ own modification [2,3] to perform the mandibular asymmetry correction surgery procedure were:(1)Patients with an asymmetrical mandible without any growth in the affected condyle in the past or at the time of the study;(2)Laterognathia without any symptoms of condylar hyperplasia (CH) or in the past according to the Wolford and Obwegeser classifications (HH/CH2);(3)Clinical records from outside the authors’ data that could have resulted in a lack of standardization or access to full clinical patient records from other institutions;(4)Mandibular asymmetry related to pathology outside of the joint, for example, tumors and pseudotumors.

Study limitations included: (1) a low number of participants in full mandible overgrowth correction related to the rarity of the disease; (2) a low number of patients qualified for additional surgery based on the scope of overgrowth that had been not sufficiently treated with standard condylectomy and orthognathic surgery; (3) patient unwillingness for any additional surgery for a definitive symmetry correction; (4) no evaluation of lateral patients profile, because of its great individuality and soft tissue changes, which is a topic for authors’ further studies; (5) a great variety of bone overgrowths and dentoalveolar discrepancies in each individual case.

A preoperative radiologically necessary measurement evaluation of the patient may be performed with RadiAnt Freeware Dicom Viewer (Medixant, Poland—Privat License). An extended radiological assessment should be delivered in patients with visible asymmetry symptoms of hyperplasia/elongation in both radiological and preclinical data using the Department of Radiology Software- GE-Discovery 750HD scanner (General Electric Healthcare, Milwaukee, USA with a dedicated workstation (Advantage Windows 4.6, General Electric Healthcare, Milwaukee, WI, USA)). The extent of the surgical approach in all cases was strictly measured on preclinical routine facial photographs and radiological studies (CT/LDCT, lateral-cephalogram, and panoramic X-ray). A detailed comparison between the excess of asymmetrical bone overgrowth and the patients’ expectations from treatment is essential.

At the first planning step, the authors proposed the investigation and measurement of the degree of bone overgrowth, using anatomical-topographical features consisting of known index measurements (IMs): the preoperative anatomical landmarks (Go-Go), (Go(Right)-Gn), (Go(Left)-Gn), and (Me–Gn) shown in Table 1. The used anatomical landmarks are commonly known [18,19,20,29,30,31,32,33]. Anthropometric points in this paper refer to ARP and consist of the following: gonion = Go refers to the most posteroinferior part of the mandibular angle; gnathion = Gn is located perpendicular on the mandibular symphysis midway between the pogonion and menton; menton = Me is the lowest point on the mandibular symphysis [20,21,22].

Likewise, anatomical reference points (ARP) measuring the degree of asymmetrical bone overgrowth should be evaluated in vertical/horizontal distances and measured based on four index values:(1)Mandibular teeth-roots/apex: MIB;(2)The distance between the roots and the superior part of the mandibular canal;(3)MIB—inferior mandibular part of the canal;(4)The distance between F0 (left: blue; right: red) and C (mandibular midline: yellow), measured between vertical and horizontal dimensions proposed by study authors.

Lines were drawn between the IM and ARP, in either the LDCT-3D, RTG/panoramic and lateral cephalogram, or others, and the orientation marks and anatomical sites with bone overgrowth and asymmetry were set. Each reference point can be easily identified, modified, and used in the presented proposal.

Secondly, the lower border of the mandible was outlined, and a vertical line from the tip of the canine, premolars, and molars was drawn perpendicular to the outlined marginal baseline. The collected radiological (panoramic) and tomographic (CT/LDCT) data were converted and archived. The horizontal volume of the bone from the tip of the roots toward the mandibular base was measured (Figure 1). Additional landmarks were set on the course of the mandibular canal to determine the distances between the roots, the mandibular base, and the mandibular canal itself.

## 3. Results

The degree of asymmetry and bone overgrowth varies in each case individually (Figure 2). A classic panoramic radiograph is sufficient to plan the scope of surgery, but because of the natural object enlargement in size by about 15–25%, any detailed measurements are troublesome. Despite this magnification, panoramic visualization enables good estimation of the degree of oversized mandibular bone, which can later be accurately measured on CT/LDCT radiographs. The authors present their proposal for approaches, the usage of which depends significantly on the anatomical-radiological distances (ARD) and proportions. All are based on the six most commonly found situations in CT/CBCT-RTG. Based on the following anatomical-topographical measurements, a protocol for surgical intervention was prepared. Perhaps the presented method of bone measurements will be a valuable study for future surgeries as condylar hyperplasia-related treatment remains without any particular guidelines and remains exceedingly individual. The detailed proposition of approaches based on the author’s proposal and own used anatomical indexes are presented in Table 2. After a detailed radiological data analysis on the RTG/panoramic and lateral cephalogram or LDCT/CT, the following anatomical-radiological proportions should be included for surgical planning:

### 3.1. Mandibular Canal

The distance between the mandibular canal and the inferior mandibular border—at least 5 mm of vertical bone overgrowth (Table 2)—could indicate the usage of marginectomy. The MIB cut (approx. 5–10 mm) is performed just under the inferior border of the mandibular canal. The degree of bone excised is measured by IM and ARP (Figure 1). When the vertical bone height is limited (<5 mm) due to a low-set mandibular canal, approach three should be used. On the other hand, marginectomy or marginectomy-swing could be performed if there is ≥7 mm of vertical bone overgrowth, and the mandibular canal is positioned ≥5 mm from the MIB. While a single cut might damage the mandibular canal, altering it with two, three, or more cuts can reduce this damage due to the curved cut line. An indication for this approach is a low-set mandibular canal with irregular placement along with the MIB, according to index measurements of teeth apices F0–F5 (Figure 1; Table 1 and Table 2). The suggested measurements in preoperative radiographs on the degree of bone overgrowth influence the used surgical approaches in the authors’ proposed perspective.

### 3.2. Gonial Angle

The length between Go-Gn/Go-Gn and the vertical bone volume relation decreases the bone height on the opposite healthy side—the critical point of this technique is the distance between the Go-Go/Go-Gn position and the value of the F0-C bone index. In this case, approach 1C can be used with or without corrective angulotomy/anguloplasty, which reduces the angled volume on the affected side in vertical and horizontal dimensions. If both Go (Right) and Go (Left) are situated on the same horizontal line, no major surgery should be planned, unless performing a condylectomy might relocate the Go point on the affected side to a higher position afterward. Therefore, careful planning in the horizontal plane and the F5: Go measurement can help estimate whether any additional mandibular angle surgery should be planned (Figure 1 and Figure 2; Table 1). This was used in some of the authors’ cases where there was no maxillary bite-plane deviation, and no chin bone asymmetry was present.

### 3.3. Chin Symmetry

The chin area position in IM and ARP (Figure 1)—severe chin area asymmetry—could be treated with the surgical approach suggested by Ferguson, while smaller ones could be successfully treated conservatively by drilling and bone reshaping or with other proposed methods of marginectomy. IM is used when the distance between C and F0 is equal (1:1); when C < F0 in both the vertical and horizontal dimensions, then a leveling and/or corrective genioplasty is used by the authors (Figure 1; Table 2). After the inferior dental neurovascular bundle is wholly free and retracted, bone correction and reshaping of the new inferior alveolar nerve canal is performed. Instead of a leveling genioplasty, the remnants of the overgrowth are smoothed with surgical drills. In the authors’ system, the primary key factor is total chin correction, which is necessary in cases of severe overgrowth; when F0 > C and Go-Gn>, there is >7 mm of vertical bone overgrowth, and the mandibular canal is positioned <5 mm from the MIB. The presented measurements are novel and not used in any previous reports and should be considered an additional tool for surgery planning.

### 3.4. Low-Dose Computer Tomography

Both panoramic radiographs and LDCT evaluations are important for any surgery planning. There is a possibility to estimate the shape and position of the mandibular canal while tracing its course on 3D evaluation and then estimating the degree of surgery in MIB (Figure 3 and Figure 4). Secondly, based on the studied patients’ data, the authors also include that the position of the mental foramina, chin deviation, and Go reference points described herein are valuable reference points for future surgical planning. Another important feature possible to estimate on 3D-LDCT is the scope of mandibular basis overgrowth in three dimensions.

### 3.5. Maxillary Secondary Deformities

Maxillary bite-plane deviation—if this is present, in the authors’ system, it should be treated with orthognathic surgery and a full osteotomy protocol. Special exclusion from the following could include: a patient’s unwillingness to undergo total osteotomy or small bite-plane deviations when masticating are not decreased, which may serve as contraindications for the procedures presented herein. In most reviewed cases, a surgical camouflage was used after the removal of the pathological growth in the condyle. All of the above-mentioned points underline that major bone discrepancies require additional procedures as perfect symmetry is difficult to achieve and major all-in-one osteotomies with swelling that progresses over time decrease the overall perception of the surgery, which the authors tried to explain in their own material and outcomes.

### 3.6. Post-Surgical Key Points

As occlusion in each patient with CH2/HH is greatly individual, an early orthodontic treatment soon after a surgical procedure is essential (preferably within 24–48 h post-surgery). In cases of a condylectomy, some premature contacts must be corrected. Camouflage treatment focuses on teeth position correction in the dental arch, along with facial/dental midline and proper occlusal cant restoration. In cases of full osteotomy protocols, all patients require strict and well-established orthodontic and surgical planning. In cases of asymmetric mandible correction, the orthodontic treatment is essential to maintain not only teeth symmetry but also jawline shape and to increase teeth and bite stability in decreasing the potential relapse related to possible muscle dysfunction.

### 3.7. 2D versus 3D Surgical Planning

The usage of a 2D-panoramic radiograph is helpful. More important measurements nowadays are made on 3D-virtual planning models (Figure 5). 3D planning might improve surgeons’ insight and the scope of bone evaluation. Because of great improvements in surgical planning, not only 3D-CT/LDCT evaluation is helpful. The usage of 3D stereolithographic models is an alternative method, where surgical cuts and plate bending might be quite easily made before surgery to ensure that the selected surgical approach is the most adequate of all. In cases of asymmetry visualized in patients in face view after orthognathic, surgical, and orthodontic treatment, some detailed total mandibular symmetry correction protocols are needed. Direct 3D-LDCT planning based on the used anatomical reference points presented herein is helpful in restoring facial balance and mandibular symmetry (Figure 6 and Figure 7). The authors’ future studies will focus on the 3D evaluation of facial skeleton, soft tissue changes in CT/MR, and lateral profile alterations; because of that, any other data are excluded from this study.

## 4. Discussion

The authors hypothesize that surgical planning in each case of condylar hyperplasia depends on the severity of factors. The total correction of mandibular asymmetrical deformities remains difficult. Each surgery focuses on the restoration of facial balance, proper bite, and occlusion, and it can be achieved in various methods [29,30,31,32]. The scope of secondary mandibular body deformities related to their overgrowths can be treated in a variety of methods. The used landmarks and anatomical proportions presented herein may also be useful. The authors’ proposal for a treatment protocol and indications for the surgical approach was based on individual bone measurements. Because most asymmetry cases are determined by different factors, including the patients’ willingness to undergo extensive surgeries at a certain age or perhaps only surgical camouflage instead of more invasive approaches, the authors tried to set some basis, indications, and contraindications for the procedures used to achieve a definitive mandibular asymmetry correction in preclinical radiological planning.

Other proposed methods suggested by Blair and Schneider [4] and Jensen [5] were milestones in correcting excess mandibular bone. The position of the inferior mandibular neurovascular bundle was a limitation for surgery because of the risk of injury or even complete damage resulting in a lack of function. Nowadays, surgeons are not limited to only using hand saws. Still, more gentle and precise instruments and techniques can be used, such as piezosurgery, which can quickly and safely cut bone in the desired places with a very low likelihood of inferior alveolar nerve damage [12,13,14,15]. Bone chisels and special retractors can be useful as all of these procedures are carried out intra-orally and require both precision and attention to the safety of vital structures, such as nerves and arteries [5,14].

All of the points, as mentioned above, underline that significant bone discrepancies require additional procedures for achieving better symmetry. Perfect symmetry is difficult to achieve [1,2,3]. When measurements have been estimated based on horizontal and vertical lines and anatomical landmarks, it is quite easy to determine the surgical margin. Ongkosuwito et al.’s study on 2D radiographs indicate that they are also quite useful [20]. As most asymmetry is found at the chin area and/or its mandibular basis with or without mandibular angle involvement, one of the approaches presented below can be used. A low-positioned mandibular canal determined whether the marginectomy should be one cut or two to three cuts to minimize damage of the inferior alveolar bundle. According to Walters et al.’s study, the degree of mandibular downward rotation also affects the scope of asymmetry [33]. It is quite obvious while planning for a marginectomy when there is a great disproportion between the height levels of both mandibular bases, comparing the health and contralateral overgrowth side.

On the other hand, if the condylectomy is to be performed at the same time as a marginectomy, the degree of open bite on the contra-lateral side influences the degree of affected/hyperplastic condyle excision. Due to the degree of resection and the formation of a new bony stump, mandibular auto-rotation is noted. In addition, the degree of bone excised from the lower mandibular base could be minimal, or none, or perhaps a marginectomy swing-approach can be used [3,7,21,24,25,26,27,28].

In the authors’ proposed protocol, based on the relationships of the above-mentioned anatomical and topographical reference points and the excess of overgrowth/asymmetry, a series of six different techniques can be used: a corrective marginectomy, a marginectomy swing approach, a three-cut marginectomy, the Modified Ferguson approach without leveling genioplasty, the Ferguson approach for total asymmetry correction with leveling genioplasty, and bone drilling and chiseling. The authors would like to highlight the usage of the anatomical and topographic reference points used herein.

We measured the values of bone disproportions on standard panoramic radiographs and low-dose computed tomography (LDCT-3D) and evaluated the amount of excess. Others focused on strict CT-3D and CT measurements for a total symmetry correction, describing that simultaneously performed condylectomy and/or orthognathic surgery does not use such anatomical reference points [25,26,27,28,29,30,31,32]. It seems that no other similar way of using anatomical-radiological proportions and landmarks has been described in the literature. The technique presented here might perhaps establish a new background for surgical planning. Combining anatomical and cephalometric landmarks help in measuring the differences and the most important relationships between structures in the study, namely, the MIB, the mandibular canal, teeth apices, the gonion, the gnathion, and the mental foramen. This diagnostic protocol helps in evaluating the degree of bone overgrowth and mandibular angulation, which sets the basis for using several of the presented surgical techniques [21,22]. Most cases of mandibular asymmetry require individual considerations [3,6,14,18,19,20]. The surgical approaches mentioned herein were used according to the accompanying radiological anatomical-topographical protocol, which is fast and straightforward. The new reference points used turned out to be very reasonable and effective in setting the basis for this classification and the surgical proposal for total mandibular asymmetry correction. A condylectomy is mandatory in all cases of condyle head pathological growth, except with a lack of growth [1,2,3,4,5,6]. The usage of any known modification of condylectomy (low, medium, high, reshaping, total, or proportional) is mostly related to either the scope of condyle overgrowth, growth factor/vector, surgeons’ preference, the scope of coexisting dentoalveolar discrepancies, or vertical/horizontal mandible ramus diameters [1,2,10,11,14,27,30,32].

The indications and selection of techniques is simply the authors’ proposal. The measurements and proportions used are helpful, mainly because the extent of each overgrowth is unique, which also requires the surgeon to evaluate each patient’s expectations.

A limitation of the paper is the variability in each case. Despite the individuality of each case, a critical position between anatomical points—such as the gonion, menton, gnathion, and pogonion—and their proportions indicate which corrective surgical protocol should be used. The measured distances between the MIB, the mandibular canal, and the teeth apexes set additional protocol reference points to determine the margin for surgical bone correction. The landmarks are used to describe the position, angulation, and proportions between several mandibular anthropometric points. All of the approaches presented herein selected based on these measurements could help restore full mandibular symmetry. Lippold et al.’s and Rodrigues et al.’s studies confirm that the growth component within the affected joint influences the scope of secondary mandibular overgrowth [1,8].

Some general considerations should be addressed. An increase in bone height between the inferior part of the mandibular canal and the MIB on the affected side allows for a swing toward the right side. The camouflage approach should include a variety of factors, e.g., patients may be unwilling to undergo any orthodontic treatment because of their age or socioeconomic status, or when bone asymmetry is the only problem, not bite or other factors [8,29,32]. During routine orthognathic surgery, chin area drilling and polishing are used instead of genioplasty. One indication of this procedure is a low or moderate degree of chin asymmetry. Total orthodontic preparation for surgery is mandatory. The degree of bone overgrowth is not essential, because when an entire orthognathic surgery protocol is used, the MIB is reduced and corrected during BSSO. Leveling genioplasty was first described and proposed by Ferguson [6]. The authors fully agree with this proposed approach, though it ought to be reserved for severe cases of overgrowth when a total surgical protocol of asymmetry correction is used. On the other hand, it should never be used for camouflage cases, but should always be related to MIB reduction and neuro-vascular bundle repositioning. Finally, bone modeling with the use of a hand saw, piezoelectric devices, and surgical burrs can also be used. It is a very conservative, but still surgical, approach that focuses on small overgrowths. This technique is very well-known and often used, so it is not necessary to describe it here [7,8,9,10,11,12,13,14,15].

Various degrees of mandibular and skeletal asymmetry caused by condylar hyperplasia and its overgrowth, treated with different surgical approaches for mandibular asymmetry correction, have been presented [3,6,14,18,19,20,25,26,27,28]. A preoperative patient assessment is required, with detailed planning and a decision about the volume of bone to be reduced or corrected during mandibular surgery. Nowadays, a complete 3D-CT/LDCT evaluation with a comparison of standard radiograms is considered to be the gold standard [21,22,30]. Patients who required orthodontic preparation and treatment had not only increased malocclusion but also significant disproportion in the skeletal bones, which required full osteotomy protocols [23,25,29,31]. The authors conclude that orthodontic treatment is not mandatory to improve mandibular symmetry, though it nevertheless enables more stable and functional outcomes in terms of bite and tooth stability. Some authors advise orthodontic treatment in almost every case [3,9,11]. Secondly, at the beginning of treatment, each surgeon should ask the patient if a full orthodontic surgical approach or perhaps a camouflage protocol is desired.

Simultaneous maxillary and mandibular osteotomies should only be performed in patients with severe skeletal malocclusion with deviations in both occlusal planes after proper orthodontic preparation, which is the treatment of choice according to most authors [1,2,3,4,5,6,7,8,9]. Lefort I osteotomy with maxillary rotation and asymmetry correction and proper bite plane restoration was used in each case where maxillary deviation was noted. This approach seems reasonable because a full orthognathic protocol enables good, stable, and accurate results [12,15,16,17,18,19,20]. Mandibular hyperplasia and its enlargement in bone size and volume are troublesome matters [1,2,3,4,5]. Most corrective jaw surgeries focus on orthognathic surgery consisting of Lefort I maxillary osteotomy and BSSO of the mandible, performed in a classic manner or with various modifications [15,16,17,23,24,25,26,27,28,29,30]. Sembronio et al.’s studies seem to confirm the authors’ thesis that advanced osteotomy protocols grant more stable final outcomes [32,33,34,35].

The most reasonable surgical approach consisted of a two-stage surgery in dentofacial discrepancies: first, a condylectomy; then, BSSO with various modalities (marginectomy, swing, or Ferguson) when the patient is willing to undergo further procedures necessary after a detailed orthodontic and scheduled procedure [6,7,11,13,14,15,16,17,18]. The authors conclude that in cases where the patient does not require or is unwilling to undergo any appropriate orthodontic treatment, a mandibular swing procedure with additional bone grafting (where necessary, from healthy mandibular bone overgrowth or other bone from the patient) on the contra-lateral side can be useful in restoring mandibular balance and improving bone coverage. On the other hand, a slow progressive growth on the condyle also resulting in the co-existing maxillary rotation should always be an indication for orthognathic surgery [1,2,15,16,17].

The authors indicate that a modification of the classic, horizontally performed, one-cut mandibular marginectomy—and changes consisting of two, three, or more cuts—can be used to reshape the overgrown mandibular base on its hyperplastic side. Distances measured with ARP: IM help to indicate the degree of bone resection (Table 2). It could also be used to reshape the mandibular angle on the hyperplastic side of the mandible or can be used with simultaneously performed anguloplasty and/or angulotomy—if the mandibular angle is enlarged and more square-shaped. Furthermore, this same approach can be used to avoid damage to the inferior alveolar nerve in its canal with the multiple horizontal cuts, while a single one could damage the nerve [3,4,5,6]. The tooth apices: MIB index values help in the possibility of lower border modeling estimation (Figure 1).

Some authors conclude that when the chin slightly deviates, then bone drilling/smoothing after an extended sagittal osteotomy is enough to improve esthetics and to minimize surgery when the chin is not fully involved. This finding is related to the low post-op growth factor in the affected condyle. Standard titanium mini plates with or without bicortical screws (18–22 mm) should be used for additional stability of the bony fragments’ position [15]. On the other hand, one bicortical screw placed in the middle of an extended osteotomy fragment is enough to stabilize the bone. Intramaxillary fixation (IMF) devices helped to secure the mandible after joint surgery.

The authors conclude that a very severe mandibular overgrowth with increased chin involvement should be a mandatory indication for the classic Ferguson approach with a surgical reduction in the mandibular angle on the affected side [7,13]. The main reason for this is the increased distance and height of the F0:F1 and F0:C:F0 values in this study (Figure 1). The authors fully agree with Ferguson’s approach, which is confirmed in the author’s measurements based on the suggested ARP: IM values and the distances between reference points in the presented study radiographs. Two different cases of one-sided mandibular overgrowths in hemimandibular hyperplasia are drawn and explained in Figure 2.

In some cases, in addition to a standard osteotomy, an additional genioplasty can be used if necessary. In the modified approach for total mandibular correction suggested by Ferguson [6], a corrective leveling genioplasty was a crucial factor in achieving full symmetry. In contrast, in the authors’ opinion, a few different approaches were used with a great overall outcome as well. Still, a leveling genioplasty should be considered mandatory in extensive bone overgrowth where the methods presented herein are insufficient, such as the proposed protocol in Table 1. CT/RTG and CBCT/3D studies [21,22,33,34,35] should be carried out, especially in the estimation of the position, height, and length of the mandibular inferior margin [7,13,14,15,16,17].

Before this paper, no attempt at describing possible vital points for a surgical protocol in total asymmetry correction had been presented or described.

In some cases, the surgeon’s clinical judgment during surgery and a comparison with the patient’s facial photographs should determine the amount of bone removed [14,15,16,21,22]. A calibrated ruler or surgical caliper is also essential for measuring the intra-operative dimensions between anatomical/cephalometric reference points. Virtual preoperative planning based on the patient’s radiographs and facial photographs is also useful [15,17,21,22].

Therefore, the authors’ perspective on the subject of restoring full mandibular symmetry is presented as clearly as possible. This study is perhaps the first attempt at describing a protocol based on specific anatomical-topographical references on preclinical radiographs.

## 5. Conclusions

A significant outcome from the protocol presented herein on the studied anatomical-topographical reference points and their proportions is instrumental in planning the degree of surgical approach in all of the known and presented strategies. Furthermore, the study presents the authors’ philosophy on total asymmetry correction planning based on the most commonly found radiological-anatomical situations in asymmetrical patients’ radiographs. The author’s idea of ARP and IM reference points could improve overall surgical planning. The Go-Gn diameter is essential for full symmetry from the angle to the chin midline. In severe chin overgrowth correction, when F0 > C and Go-Gn>, and there is >7 mm of vertical bone overgrowth, Ferguson’s approach should be used. The surgical procedures presented herein can be used successfully in all cases of mandibular asymmetry, resulting most commonly from hemimandibular hyperplasia, and should improve patient’s expectations.

## Figures and Tables

**Figure 1 ijerph-19-10005-f001:**
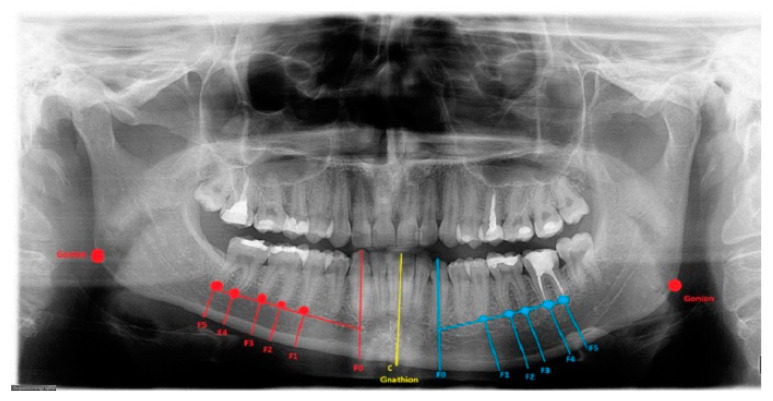
Mild form of condylar hyperplasia. Red and blue lines marked F0–F5 help to visualize the distances between the tooth apex, the mandibular canal, and the mandibular base. Gonion and gnathion landmarks help to define right/left bone proportions and lengths. The yellow midline always marks the teeth and the mandibular midline. The degree of bone overgrowth and mandible angle shape and volume can be easily estimated.

**Figure 2 ijerph-19-10005-f002:**
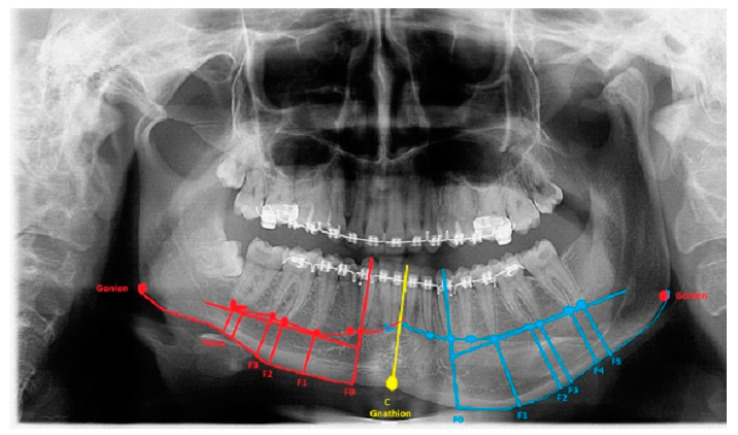
Severe form of condylar hyperplasia.

**Figure 3 ijerph-19-10005-f003:**
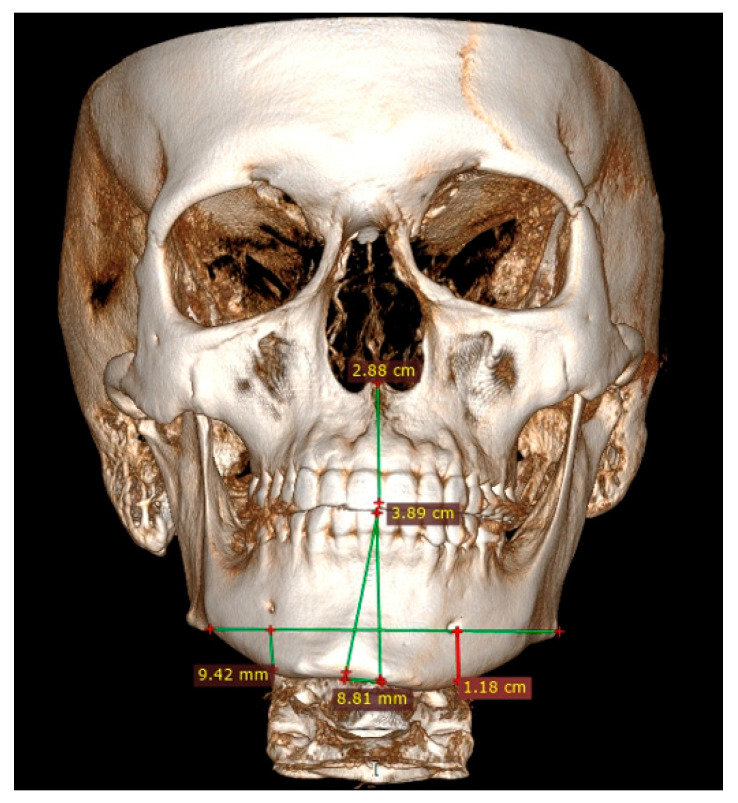
LDCT—evaluation in low-dose CT focused on central symmetry line and differences in measurements between mental foramina, MIB, chin position, and mandibular angle contour.

**Figure 4 ijerph-19-10005-f004:**
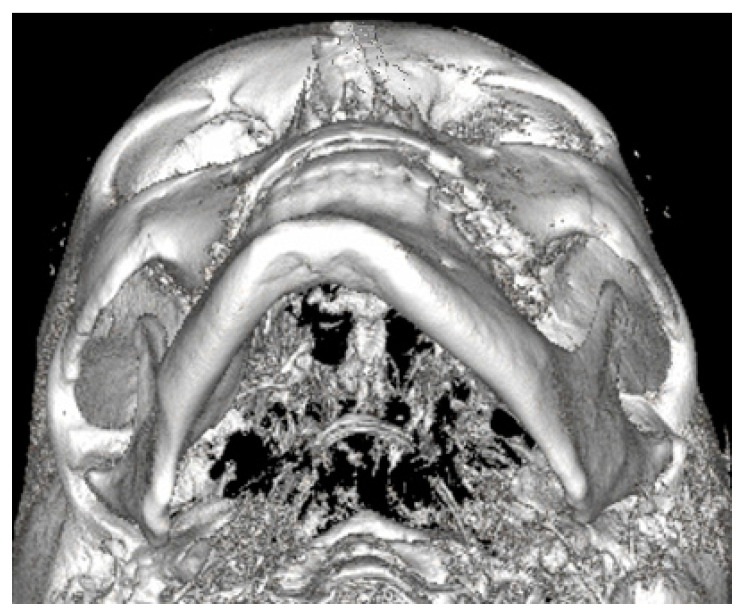
LDCT semi-axial view on the deviated mandible. Except for chin deviation, elongation of left mandibular basis, and slight Go-point asymmetry, the mandibular body is not enlarged nor presents overgrowth in a significant diameter.

**Figure 5 ijerph-19-10005-f005:**
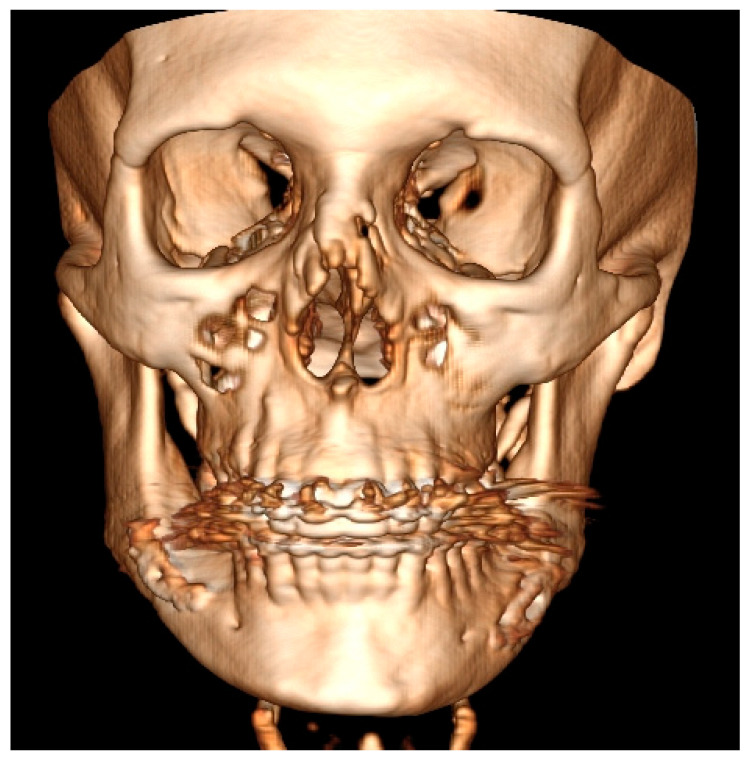
Patient CT evaluation after BSSO. Presented frontal scan underlines still present asymmetry in the left mandibular basis, which was not fully camouflaged after BSSO and required further surgical approaches for definitive mandibular symmetry correction.

**Figure 6 ijerph-19-10005-f006:**
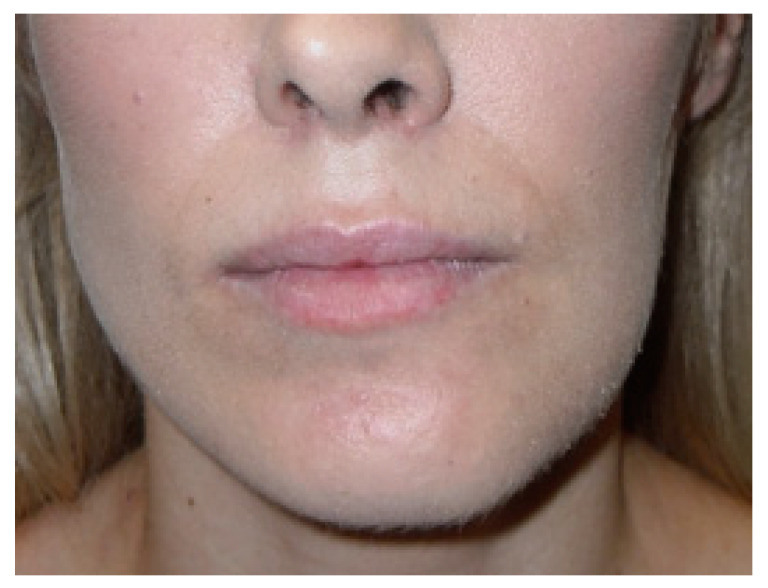
Patient with visual left mandibular basis overgrowth in a case of inactive UCH2. A mild form of facial asymmetry.

**Figure 7 ijerph-19-10005-f007:**
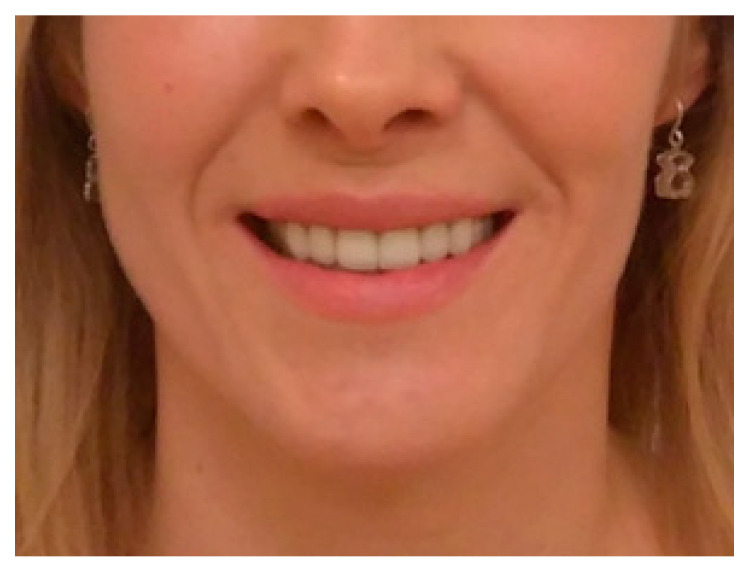
Same patient after a left mandibular basis marginectomy with bone modeling after 8 months.

**Table 1 ijerph-19-10005-t001:** Proposed usage of anatomical reference points in surgery planning.

**Anatomical reference points**	(Go-Go) (Go(Right)-Gn) (Go(Left)-Gn) (Me–Gn)	See reference point distances between: (1) mandibular teeth and roots/apex: MIB; (2) roots and superior part of mandibular canal; (3) MIB and inferior mandibular part of the canal; (4) F0 (**left-blue**/**right-red**) and C (**mandibular midline-yellow**) measured between vertical and horizontal dimensions.

Abbreviations: MIB—mandibular inferior border (mandibular basis); mm—millimeters (1 cm = 10 mm). Anatomical landmarks: (Gonion = Go, gnathion = Gn, menton = Me)-index:- (Go-Go)/(Go(Right)-Gn)/(Go(Left)-Gn)/(Me–Gn) anatomical landmarks [18,19,20]; F0—vertical line drawn from mandibular canine (**left-blue**/**right-red**) toward MIB; C—vertical mandibular midline between central mandibular incisors toward MIB (**yellow**).

**Table 2 ijerph-19-10005-t002:** Authors’ proposal for mandibular correction protocol.

Five Key Factors	Yes/No	Procedure	Sufficient Yes/No	Additional Procedure
**Maxillary bite plane rotation/asymmetry**	Yes No	Orthognathic surgery + orthodontic preparation Surgical camouflage with marginectomy/mandibular-swing or a single condylectomy	No No	Ferguson or modified Ferguson technique with/without marginectomy Bone chiseling or bone grafting
**Active growth in condyle**	Yes No	Condylectomy Orthognathic surgery or camouflage surgery	No No	Mandibular bone remodeling a/s
**Mandibular overgrowth**	Yes No	Marginectomy or mandibular-swing or modified-Ferguson Technique with BSSO or classic-Ferguson technique suggested by Ferguson Marginectomy or one-two-three cuts with or without angulotomy/anguloplasty	Yes No	Mandibular bone remodeling (ex., chiseling) and/or bone grafting
**Skeletal open bite**	Yes No	Orthodontic treatment Surgical camouflage with or without marginectomy	No No	Orthognathic surgery Mandibular bone remodeling (ex., chiseling) and/or bone grafting
**Dento-alveolar discrepancies**	Yes No	Orthodontic treatment Surgical camouflage	No	a/s
**Mandibular canal position**	Normal Low position: <5 mm from MIB High position: >8 mm from MIB		BSSO + bone remodeling (ex., chiseling) BSSO +/-Ferguson approach Marginectomy or bone-swing MIB	
**Degree of bone overgrowth**	None Small: <4 mm Medium: 5–8 mm and High: >9 mm		Orthognathic surgery Bone remodeling on MIB/chin area Marginectomy or mandibular-swing or modified-Ferguson Technique with BSSO or classic-Ferguson technique suggested by Ferguson; Marginectomy or one-two-three cuts with or without angulotomy/anguloplasty	
**Occurrence of open bite**	Yes; Small 2–3 teeth Medium 4–5 teeth Total-one sided No		Condylectomy-with mandible auto-rotation BSSO + orthodontic treatment Orthognathic surgery and/or Ferguson app. and/or Camouflage surgery, marginectomy, BSSO Condylectomy with camouflage surgery-bone drilling/marginectomy and/or swing-app	

Abbreviations: MIB—mandibular inferior border (mandibular basis); mm—millimeters (1 cm = 10 mm); a/s—as mentioned above; ex. = example; BSSO—bilateral sagittal split osteotomy of the mandible; Ferguson technique—(modified technique presented in: J Cranio-Maxillo-Fac Surg 33: 150-7, 2005 by Ferguson JW). Anatomical landmarks: (Gonion, gnathion, and menton)-index:- (Go-Go)/(Go(Right)-Gn)/(Go(Left)-Gn)/(Me–Gn) anatomical landmarks [18,19,20]; app—approach; mandible auto-rotation—depends on degree of excised condylar head and degree of open bite on affected side. ARP—anatomical reference points; IM—index measurements.

## Data Availability

The datasets used and/or analyzed during the current study are available from the corresponding author on reasonable request.

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
