# Peer review of "Methods of Definitive Correction of Mandibular Deformity in Hemimandibular Hyperplasia Based on Radiological, Anatomical, and Topographical Measurements—Proposition of Author’s Own Protocol"

_ijerph, 2022, doi:10.3390/ijerph191610005_

Round 1

Reviewer 1 Report

This is a very interesting paper detailing a proposed protocol for surgical treatment of mandibular asymmetry.

The paper is original and could be of great interest to readers. 

I have a few minor comments:

Abbreviations should be explained when first used.

The discussions section is long and derivative, please shorten it where possible.

Did the authors take into account the profile of the patient? It can significantly alter the final outcome of surgery.

The conclusions must be rewritten in a more concise manner.

Author Response

Hello. Thank you for the review. All questions are attached and discussed.

Reviewer 2 Report

I had a lot of expectations after reading the title of the article, because it seemed to be an interesting topic. However, the title does not reflect very well the content of the article. I recommend to reconsider the title.

First of all, the english language and style must be serious improved because it is very hard for the reader to follow the topic. I recommend that the article be rewritten and checked by an english native speaker.

INTRODUCTION section:

-line 46-47 must be moved at the end of the section, not in the first paragraph

-the citations are chaotic, must follow the guidelines of the journal. It is not possible to have 1,7,14,23-must be arranged in ascending order

-the introduction is not focused and correlated with the title. Beside this aspect, must be shortened.

-the abbreviations must be explained

-IAN palsy is not correct from my point of view, IAN being a sensitive nerve, not a motor one. Maybe you wanted to discuss about the facial nerve paralysis

MATERIALS AND METHOD section:

-please state the time period when the study was conducted

-the number of the ethic committee must be stated here

-the study limitations are written twice 

-in this section the surgery part is not debated at all. Why?

DISCUSSION section:

-this section has a lot of informations that can/must be moved in the introduction section. This section must be shortened 

-why table 2 is not described in the materials and method section? From my point of view, it does not belong to the discussion section

-,,the authors measured”- use our team/we measured etc

The idea of the study is very interesting but the scientific article is not well written at all. 

Author Response

(The authors gave the same response as above.)

Reviewer 3 Report

The present research paper approaches a very interesting topic and provides valuable insight into the field. Also, it is very well written and organized!

There are still a few aspects that can be improved.

The presented evaluation protocol presents several tooth landmarks which has two flaws: first of all, it assumes that teeth on both sides of the mandible are symmetrical which due to physiological and sometimes pathological conditions is not always true. Also, some of these teeth might be missing and extended edentations  will become very problematic for the measurements. Please find solutions for these issues.

Patients with asymmetries present themselves to the doctor's office for both aesthetical and functional reasons. Although the asymmetry is given by the bone, the soft tissues are actually the ones that are visible. Little is the aspect of soft tissue response after surgery discussed. 

A 3D printed model can be very helpful in these cases, in order to visualize the preoperative situation and also to plan and simulate the surgery. This technique was not discussed or compared to the proposed protocol.  

Although the provided references are adequate and relevant, not to many papers were cited from the last 5 years. Please check whether important recent papers were not ignored. 

Overall, this is a very good and interesting paper! Congratulations! Please address the mentioned aspects in order to make it more relevant.

Author Response

(The authors gave the same response as above.)

Reviewer 4 Report

The major flaw of  this study is it's retrospective character. The number of cases is too small to draw any conclusions (12 cases and 8 solutions as in a case series). There are no objective data confirming proposed protocol, e.g. comparison of measurements taken before and after surgery, evaulation  of aesthetic and functional outcome, statistical analysis of obtained results which would confirm chosen protocol. Due to these serious problems, this paper does not meet requirements for a medical article fit for publication.

In the future, please consider the proper form of the manuscript with concise introduction (some parts of your introduction should be placed in the "discussion section" ), methodology should include clear inclusion/ exclusion criteria, protocol of the study with clearly stated diagnostic methods and measurements taken but without numbers, which sould be placed in the "results" section. Morover, methods of evaluation of the results should be presented along with chosen methods of statistical analysis. In the "results section" present your results. One can not propose extensive surgery as own protocol when it is based on 1-2 cases! This must be supported by proper number of treated patients and sound evaluation of the obtained results. The references ought to be presented in the  order of citation in your text.

Author Response

(The authors gave the same response as above.)

Reviewer 5 Report

The authors present their protocol for correction of mandibular deformity in hemimandibular hyperplasia based on anatomical-topgraphical measurements.

The manuscript would benefit from one or two clinical cases of successful treatment of patients with hemimandibular hyperplasia using the presented protocol.

The authors should discuss the value of 3D imaging and virtual planning in the treatment of patients with hemimandibular hyperplasia in comparison to their 2D protocol.

Author Response

(The authors gave the same response as above.)

Round 2

Reviewer 2 Report

Thank you for the revised version.

My previous request ,,he citations are chaotic, must follow the guidelines of the journal. It is not possible to have 1,7,14,23-must be arranged in ascending order

Response 4: . –references are re-arranged and clear. This request was not resolved. Look at line 61, where you have cited 1 and after is 7. Please revise!

All other modifications were resolved according to my request and I would like to thank the authors.

Author Response

Point 1. My previous request ,,he citations are chaotic, must follow the guidelines of the journal. It is not possible to have 1,7,14,23-must be arranged in ascending order

Response 1: Please provide your response for Point 1. (in red) – Thank you.  The refferences are corrected in order : known literature [1,7,9-14,23-32]. Numbers 23-32 are deletd and improved numbers on same topic and adressed as 1,7,9-18. Thank you

Point 2 Response 4: . –references are re-arranged and clear. This request was not resolved. Look at line 61, where you have cited 1 and after is 7. Please revise!.

Response 2: . Please provide your response for Point 1. (in red) – Thank you.  Please read the text. First citation described refrences from 1-8, later all references are cited 1,7 and 9-14. It looks quite clear and in order.

[…]”Condylar hyperplasia is related to one-sided abnormal pathological overgrowth of the mandible, related to condyle head increased and prolonged growth in time [1-83]. Its etiology remains not fully known, while occurrence time, gender-related factors, growth components, and severity are widely discussed in known literature [1,7,9-18,23-32]….[…]”

Point 3 All other modifications were resolved according to my request and I would like to thank the authors.

Response 3: . Please provide your response for Point 1. (in red) – Thank you. All the wise comments and reviewer remarks are just remarkable and precious! I would like to thank, In behalf of myself, and entire team, for all the valuable comments and help in improving the paper! Thank you

Reviewer 5 Report

The authors have clearly adressed the reviewer's comments. The manuscript can be accepted for publication.

Author Response

Response 1: Please provide your response for Point 1. (in red) – Thank you. I am very happy and please that its suitable for further publication. It wouldn’t be such a great paper without such a great reviewer help and comments. Thank you.  
